# The Role of *Globularia alypum* Explored Ex Vivo In Vitro on Human Colon Biopsies from Ulcerative Colitis Patients

**DOI:** 10.3390/nu15061457

**Published:** 2023-03-17

**Authors:** Najla Hajji, Ilaria Russo, Jessica Bianco, Ornella Piazza, Paola Iovino, Antonella Santonicola, Carolina Ciacci

**Affiliations:** Department of Medicine, Surgery and Dentistry, Scuola Medica Salernitana, University of Salerno, Baronissi, 84081 Salerno, Italy

**Keywords:** *Globularia alypum*, oxidative stress, ulcerative colitis, interleukin 6, COX-2, NF-κB

## Abstract

The existing literature indicates that *Globularia alypum* L. (GA) influences inflammation and oxidative stress modulation in rats and in vitro. The present study aims to investigate the effects of this plant in patients with ulcerative colitis (UC) and normal controls. In our experiments, we pretreated colon biopsies from 46 UC patients and normal controls with GA leaves aqueous extract (GAAE) used at two concentrations (50 and 100 µg/mL) for 3 h, followed by Lipopolysaccharides (from *Escherichia coli*) stimulation. We analyzed the effects on inflammation by studying the cyclo-oxygenase-2, the intercellular adhesion molecule-1, the nuclear factor kappa B, and p38 mitogen-activated protein kinase expression. Moreover, we assessed the levels of interleukin 6, the superoxide dismutase activity, and nitric oxide release in the supernatant of cultures. Our data showed that GAAE influences UC patients and normal controls for most studied markers and enzymes. These results acknowledge, with some scientific evidence, the traditional belief in the anti-inflammatory properties of GA and represent the first demonstration of its effect in a human in vitro model of inflammatory conditions.

## 1. Introduction

*Globularia alypum* L. (GA) is a Mediterranean-growing plant in the *Globulariaceae* family [1]. Historically, mainly the leaves of GA were used in traditional medicine to treat various ailments induced or accompanied by inflammation, including rheumatism, infections, gout, typhoid, intermittent fever, constipation, and diabetes [2,3,4]. Some research has demonstrated in vitro that GA has antibiotic properties and antileukemic powers [3,5]. The active principles in the flowers, leaves, and stems of GA methanol and dichloromethane extracts are supposed to antagonize the contractile response induced by different neurotransmitters [6]. More recently, GA methanol extract has shown a potent anti-oxidant effect [1,7,8], while GA petroleum ether extract showed anti-tuberculosis activity against *Mycobacterium tuberculosis* [7]. Numerous studies have demonstrated that GA is abundant in secondary metabolites, such as polyphenols and iridoids [9,10,11,12]. The aqueous GA leaves extract (GAAE) contains active compounds with androgenic qualities that enhance active spermatogenesis in mice at a daily dose of 100 mg/kg for 15 days, and so has the potential for human male infertility treatment [13]. Another known effect of GAAE is the hypoglycemic role coupled with the hypotriglyceridemic effect, demonstrated in rats by repeated oral administration [6]. GA works in the muscle and kidney by lowering lipid peroxidation and making anti-oxidant enzymes work better [8]. Previous in vivo studies showed a laxative effect of GAAE in constipated rats [14], and an anti-inflammatory effect in rats with induced ulcerative colitis (UC) by decreasing the pro-oxidant enzyme activities, in particular superoxide dismutase (SOD) [11]. Furthermore, an acute toxicity analysis demonstrated that the aqueous extracts are relatively safe because the GA LD_50_ value in rats was found to be above 14.5 g/kg [15], and a 10 g/kg dosage was not deadly in mice [13]. Despite the plant’s long history of usage as a folk treatment in the Mediterranean, there is currently only a small amount of scientific research evaluating the safety and anti-inflammatory benefits of administering the GAAE to humans.

Among human diseases causing inflammation, UC is an idiopathic chronic inflammatory disease of the gastrointestinal tract [16]. Colitis, rectal bleeding, diarrhea, stomach discomfort, and weight loss are all symptoms of UC [17,18]. Even though UC’s pathophysiology is still unclear, experimental and clinical evidence suggests that chronic intestinal inflammation may be caused by a dysfunction of the immune system, smoking, and psychological factors [19]. These factors can cause inflammatory cytokines and chemokines to be released and signaling pathways to be turned on. These include tumor necrosis factor-α, nuclear factor kappa B (NF-κB), mitogen-activated protein kinases p38 (p38 MAPK), interleukin -1β, interleukin 6 (IL-6) and 17, interferon-γ, cyclo-oxygenase-2 (COX-2), and intercellular adhesion molecule 1 (ICAM-1) [20,21]. It is possible that oxidative stress, which is linked to persistent intestinal inflammation, is a key factor in the development of the condition [22].

The current work investigates some of the known anti-inflammatory and anti-oxidant capabilities and effects of GAAE in an ex vivo in vitro model of human gut colonic explants from UC patients and controls.

## 2. Material and Methods

### 2.1. GA Collection and Extract Preparation

GA was gathered in March 2018 from the region of Boussalem (north-west Tunisia). This specimen has been cataloged as GA-ISBB-03/03/18 and is housed in the herbarium of the Higher Institute of Biotechnology of Beja at Jendouba University. For 72 h at 40 °C in an incubator, GA leaves (10%, weight/volume) were dried and then ground in an electric mixer. The plant powder was then added to distilled water, and the mixture was incubated at room temperature for 24 h with magnetic stirring. After centrifugation (10 min, 10,000× *g*), the resulting GAAE was lyophilized and kept at −80 °C [23].

### 2.2. Patients and Biopsy Culture

#### 2.2.1. Patients and Collection of Biopsies

With their agreement, we gathered data and biopsies from 31 patients with UC who underwent colonoscopy for diagnosis or routine follow-up and 15 controls who also underwent colonoscopy for hemorrhoid bleeding or irritable bowel syndrome but had no abnormal findings. Several biopsies were collected from various areas of the colon throughout the procedure. Every biopsy was cultured right after removal by placing it in a sodium chloride solution (0.09%).

#### 2.2.2. Biopsy Culture and GAAE Treatments

The cultured intestinal explants provide an ex vivo in vitro model that faithfully recapitulates the intestinal environment down to the individual cell populations and their interdependencies.

The biopsies were put villous-side up on a stainless steel mesh and then placed above the central well of an organ culture dish (Falcon, Franklin Lakes, NJ, USA). The biopsies’ slicing surfaces were brought into contact with the culture medium by adding it to each well. DMEM F12 (16 mL), fetal calf serum (3 mL), 50,000 IU penicillin, and 5000 IU streptomycin made up the culture medium. At 37 degrees Celsius, the dishes were gassed with 95% oxygen and 5% carbon dioxide in an anaerobic jar [16].

One well contained just culture medium as a negative control, four wells were pretreated with GAAE (50 and 100 µg/mL), and one well served as a positive control. After 3 h of incubation, each culture was given 1 µg/mL of *Escherichia coli*-lipopolysaccharides (EC-LPS) and left to sit overnight. We terminated all cultures after the first day. For further analysis, the tissue was flash-frozen in liquid nitrogen and kept at −80 °C.

### 2.3. Immunohistochemistry

All the cultured biopsies obtained from the 31 UC and 15 control colons were tested by immunohistochemistry. Cryostat 5 µm thick mucosal slices were created and utilized for immunological labeling. Sections were fixed in acetone for 15 min before being treated separately with the following antibodies: human anti-COX-2 (diluted 1:100 CAYMAN, chemical business, Ann Arbor, MI, USA), human anti-ICAM-1 (diluted 1:100; Santa Cruz Biotechnology, Dallas, TX, USA), Human p38 MAPK (diluted 1:200; Bioss Antibodies, Woburn, MA, USA). However, for the NF-κB staining, after being fixed in 4% formaldehyde for 15 min, sections were permeabilized for 5 min with Triton and washed three times in PBS. Then, they were incubated in block buffer (0.3% Triton, 10% BSA in PBS) for one hour. The human anti-NF-κB antibody (Bioss Antibodies, Woburn, MA, USA, diluted 1:100) was added and incubated overnight.

After the first incubation, all sections were washed three times with PBS. Alexa Fluor 488 conjugated anti-mouse IgG was added to the sections for 60 min at room temperature, and the sections were then washed three times with PBS. The tissue sections were then put in a solution of DAPI (1:1000) for 5 min, after which they were mounted with a solution of 20% glycerol in PBS. A fluorescence examination with a Nikon eclipse was used to examine the data. Object 20X and software for processing images were used to take the pictures.

### 2.4. Western Blot

Samples of the intestine were broken up in RIPA buffer in the presence of protease and phosphatase inhibitors. The Bradford (Biorad, Hercules, CA, USA) method measured the amount of proteins in the supernatant. The concentration of proteins was calculated using a standard range of BSA with known concentrations. From each biopsy, 40 μg of total proteins are adjusted with lysis buffer (RIPA), mixed with 10 μL of LAEMMLI buffer, and then denatured at 95 °C for 5 min. Ten percent of sodium dodecyl sulfate/polyacrylamide gel electrophoresis was used to put the lysates on the gel (SDS-PAGE).

The migration was performed at 80 V and subsequently at 120 V. The proteins were then transferred from the gel to a nitrocellulose membrane using a transfer cassette (Biorad, Hercules, CA, USA) and incubated for one hour at room temperature with a blocking solution (5% dehydrated milk powder dissolved in TBST). Monoclonal antibodies against COX-2, NF-κB, ICAM-1, and p38 MAPK were applied to membranes overnight at 4 °C. Following three TBST buffer washes, the membranes were treated for one hour with horseradish peroxidase-linked goat anti-mouse or anti-rabbit secondary antibodies (1:1000). The rabbit antibody-actin and lamin A/C were used as internal controls (1:1000; ABCAM, Cambridge, UK). The Chemidoc was used to visualize the findings of Western blot detection reagents (Clarity Western ECL substrate, Biorad, Hercules, CA, USA).

### 2.5. Nitric Oxide Dosage

As previously described [24], nitric oxide (NO) levels in the gas phase were measured using a Sievers NOA 280 A chemiluminescence analyzer. To liberate gaseous NO from dissolved NO and nitrite, 100 µL of culture medium samples were pumped into a nitrogen purge tube containing 1% sodium iodide in glacial acetic acid solution. The sample gas was then exposed to ozone in the reaction vessel, producing activated nitrogen dioxide, which was detected with a red-sensitive photomultiplier tube and recorded with an integrated pen recorder. Using a calibration curve developed by examining a series of sodium nitrite standards, the area under the curve for each sample was converted to picomolar NO.

### 2.6. The SOD Activity

The SOD activity was assessed using the Misra and Fridovich technique [25], which is based on SOD’s capacity to convert the superoxide anion to peroxide. At basic pH, hydrogen competes with superoxide anion for the autoxidation of epinephrine. In brief, 5 µL of the sample (biopsy lysate supernatant) was added to the bovine catalase (0.4 U/µL) buffered with carbonate/sodium bicarbonate (62.5 mM; pH 10.2). The optical density was determined at 480 nm and adjusted to zero. The reaction solution was then supplemented with epinephrine (5 mg/mL), and SOD activity was evaluated by measuring changes in absorbance every 30 s for a total duration of 5 min at 480 nm.

### 2.7. ELISA Test for IL-6 Assay

The ELISA method was used to measure the amounts of IL-6 in the supernatant, and a commercial kit was utilized for the measurement (MyBioSource, San Diego, CA, USA). The protein levels were determined by employing a microplate reader with the 450 nm wavelength setting (Tecan Sunrise RC, Tecan, Mannedorf, Switzerland). The amounts of protein were adjusted such that they were equivalent to the conventional levels of protein.

### 2.8. Statistical Analysis

Data are presented as means SEM and were carried out using Student’s *t*-test for paired and unpaired data when necessary. Values were considered statistically significant at *p* < 0.05.

## 3. Results

### 3.1. GAAE Pre-Treatment Effect in Normal Colon Biopsies upon EC-LPS Challenge

#### 3.1.1. Effect on Inflammatory Markers

##### COX-2 and ICAM-1

The expression of ICAM-1 in situ (Figure 1) was greatly enhanced by EC-LPS in normal colons (Figure 1B). However, biopsies pretreated with 50 and 100 µg/mL of GAAE before being treated with EC-LPS exhibited a considerable reduction in ICAM-1 in comparison to EC-LPS alone (Figure 1C,D). Compared to non-treated biopsies, biopsies treated simply with GAAE revealed a normal ICAM-1 expression (Figure 1E).

Appendix A summarize the results and the statistical analyses of all experiments. 

The COX-2 in situ expression is shown in Figure 2. The EC-LPS has increased COX-2 positive cells compared to the control M. Results from GAAE pre-treatment 50 µg/ml were almost like EC-LPS treatment (Figure 2C), but the dose of 100 µg/ml showed a significant decrease of positive cells (Figure 2D). Biopsies treated only with GAAE (100 µg/ml) showed almost no COX-2 positive cells (Figure 2E).

##### Effect on NF-κB and p38 MAPK

Most controls in situ (Appendix A, Figure 3B and Figure 4B) have demonstrated an upregulation of NF-κB (Figure 3) and p38 MAPK (Figure 4) following EC-LPS treatment. Both NF-κB and p38 MAPK were more strongly affected by GAAE at the higher of its two dosages (Figure 3C,D and Figure 4C,D). Biopsies cultured in the absence of treatment or with 100 µg/mL GAAE but without EC-LPS stimulation exhibited reduced NF-κB and p38 MAPK (Figure 3E and Figure 4E) expression (Figure 3A and Figure 4A). The statistical analyses of the immunofluorescence are shown in the Appendix A.

##### Effect on IL-6 Production

The EC-LPS challenge caused an increase in the levels of IL-6 in the biopsies. Still, this rise was suppressed when the colons were pretreated with either 50 or 100 µg/mL of GAAE (Figure 5). The levels of IL-6 in cultures that were simply treated with 100 µg/mL of GAAE were much lower than those in the control M (Medium).

#### 3.1.2. Effect on Oxidative Stress Markers

##### Effect on SOD Activity

After EC-LPS stimulation, there was a discernible rise in the measured enzyme activity. This activity decreased after GAAE pre-treatment, especially with 100 µg/mL(Figure 6).

##### NO Production

The EC-LPS stimulation strongly increased NO production in all biopsies (Figure 7). The pre-treatment with GAAE decreased NO production in controls, especially with 100 µg/mL. The treatment with GAAE without EC-LPS stimulation showed the same level of NO production of control cultures M.

### 3.2. GAAE Pre-Treatment Effect in Colon Biopsies from UC Patients upon EC-LPS Challenge

#### 3.2.1. Effect on Inflammatory Markers

##### Effect on COX-2 and ICAM-1 Activity

Results from the immunohistochemistry of biopsies from UC patients showed a more interesting effect of EC-LPS on COX-2 and ICAM-1 expression (Figure 8 and Figure 9), respectively. Appendix A summarize the results and the statistical analyses of the immunofluorescence. 

Western blotting results confirmed the in situ results (Figure 8F and Figure 9F) and proved the strong effect of GAAE in decreasing those markers. GAAE decreased ICAM-1 expression to 49.6% and 33.3% when combined with EC-LPS. GAAE decreased the COX-2 expression by 62.5% and 20% in combination with EC-LPS.

##### Effect on NF-κB and p38 MAPK Expression

In UC patients, NF-κB and p38 MAPK expressions were mostly higher in patients under EC-LPS stimulation (Appendix A; Figure 10B and Figure 11B). On the other hand, GAAE had a visible effect on decreasing p38 MAPK (Figure 11C), and NF-κB (Figure 10C) activity in the presence of EC-LPS and almost attenuated its activity in the absence of EC-LPS stimulation (Figure 10D). The results and the statistical analyses of these figures are presented in the Appendix A. 

The determination of the expression level of both NF-κB and p38 MAPK was also performed by Western blotting, which proved the antagonist effect of GAAE, particularly on NF-κB expression (Figure 10F and Figure 11F). The expression of NF-κB was decreased by 45.1% in the presence of GAAE alone and by 73.6% when GAAE was added with the presence of EC-LPS. Meanwhile, the expression of the p38 MAPK was decreased by 49% under GAAE treatment and by 61% in the co-presence of GAAE and EC-LPS.

##### Effect on IL-6 Production

In the biopsies derived from UC patients (Figure 12), we observed an over-expression of this cytokine compared to control patient cultures (Figure 5). A production increase was detected in most of the cultures after the EC-LPS stimulation (Figure 12, EC-LPS). GAAE pre-treatment significantly affected EC-LPS response, while GAAE alone did not significantly affect IL-6 production compared to non-treated biopsies M.

#### 3.2.2. Effect on Oxidative Stress Markers

##### Effect on SOD Activity

The SOD activity was also evaluated from the biopsies of six UC patients (Figure 13). An increase in enzyme activity was detected after EC-LPS stimulation, which was partially reduced after GAAE pre-treatment.

##### Effect on NO Production

The EC-LPS stimulation increased NO production in most cultures (Figure 14). In those cultures, the pre-treatment with GAAE decreased NO production level either in biopsies treated in the presence or absence of EC-LPS.

## 4. Discussion

The increased interest in medicinal plant extracts in treating intestinal inflammatory diseases has prompted several clinical studies assessing the potential pharmacological properties of plants, their side effects, and costs [26].

We investigated GAAE treatment in colon explants from controls, and UC patients challenged with EC-LPS, showing a significant control of the inflammation.

In our setting, normal control colon biopsy cultures showed an increase in levels of NO, SOD activity, COX-2, NF-κB, ICAM-1, p38 MAPK, and IL-6 expression upon stimulation with EC-LPS. In addition, colon biopsies from UC patients showed an increase in NF-κB and COX-2 expression under the EC-LPS effect. However, in UC, the ICAM-1 and p38 MAPK, already increased for the disease-induced inflammation, were less affected by the EC-LPS exposure than in controls. The GAAE pre-treatment remarkably decreased COX-2 and NF-κB. In addition, IL-6 levels increased with EC-LPS exposure in UC and were partially inhibited after GAAE pre-treatment.

Previous studies in healthy rats [27] and human colon biopsies taken from healthy subjects [28] revealed that EC-LPS exposure raised several inflammatory indicators, including IL-6. EC-LPS works as a switch for macrophage activation, as indicated by increased production of IL-6, NO, tumor necrosis factor-α (TNF-α), prostaglandin E2, interleukin-1β, IL-10, inducible nitric oxide synthase (iNOS), and monocyte chemoattractant protein 1 [29,30]. MAPKs and NF-κB signaling pathways were proposed to be the two major intracellular molecular pathways involved in the inflammatory cascade response to EC-LPS activation in RAW264.7 cells [30].

Our data indicate that GAAE pre-treatment reduces the activity of some of those markers in colon biopsies and further demonstrates that NF-κB is primarily involved in EC-LPS inflammatory activation. The observed high level of ICAM-1 before the UC biopsies stimulation may explain why the EC-LPS exposure had no high influence on the ICAM-1 level. On the other hand, GAAE inhibited ICAM-1 much more in normal biopsies than in UC samples. Therefore, GAAE seems to act preventatively against ICAM-1 expression more than curatively.

The NF-κB is essential for producing pro-inflammatory genes, such as IL-6 [31]. IL-6 is associated with intestinal epithelial cell proliferation [32]. Recent research suggests a relationship between chronic inflammatory disorders and IL-6 signaling [33,34]. NF-κB suppression may reduce cytokine production and affect ROS/RNS generation in inflammatory bowel disease patients, particularly during the disease’s active phase [35]. A study suggested a positive link between NOS-derived NO and IL-6, IL-17A, and IL-23 plasma levels in inflammatory bowel disease patients [36]. The radical scavenger NO may also mediate pro-oxidant actions that take over when there is inflammation or immunological activity in the gastrointestinal system [37]. Over a prolonged period, the overall quantity of NO in the inflamed gut mucosa seems significantly high [32].

In the present study, UC colon biopsy demonstrated an increase in oxidative stress, as evidenced by the up-regulation of SOD and NO production upon EC-LPS stimulation. The present findings are consistent with prior in vivo investigations in rats, utilizing GAAE as a therapy for acetic acid-induced colitis [11]. The activation of SOD is thought to protect the intestinal tissues from oxidative damage caused by inflammation and oxidative stress. The levels of SOD in the peripheral blood of patients with inflammatory bowel disease are presently being employed as a biomarker of oxidative stress [38].

Our data confirm the GAAE’s ability to scavenge ROS [10] and establish anti-oxidant enzyme levels, protecting against colon inflammation [11].

The study limitations were the small number of individuals included and the limited number of biomarkers investigated. The number of individuals was initially greater than presented here. However, we had to exclude many cases because of inadequate culture conditions or the small size of biopsies, which limited the number of tests, giving incomplete results. As per the limited number of biomarkers assessed, we chose the biomarkers directly involved in illness manifestation and activation by EC-LPS. Other biomarkers, particularly cytokines, might be implicated; however, the size of the biopsies and the amount of the culture media, together with the several repeats for each test, limited the study’s ability to cover a broader range of tests.

In conclusion, data from our pilot trial formulate the hypothesis that GAAE should be further investigated as an adjuvant therapy for UC/inflammatory bowel diseases, but more interestingly, to prevent chronic intestinal inflammation.

## Figures and Tables

**Figure 1 nutrients-15-01457-f001:**
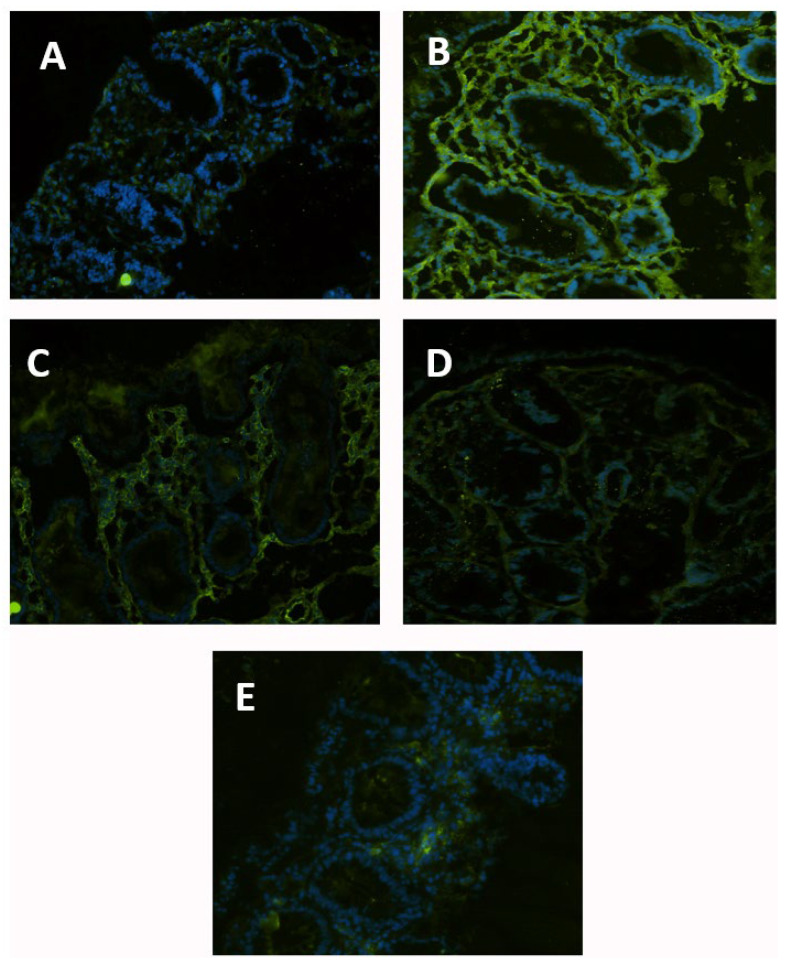
**GAAE pre-treatment effect on ICAM-1 expression in normal controls challenged with EC-LPS.** (**A**) Non-treated biopsy; (**B**) biopsy challenged with 1 µg/mL EC-LPS for 18 h; (**C**) biopsy pretreated 3 h with 50 µg/mL GAAE, then challenged with 1 µg/mL EC-LPS for 18 h; (**D**) biopsy pretreated 3 h with 100 µg/mL GAAE, then challenged with 1 µg/mL EC-LPS for 18 h; (**E**) biopsy pretreated only with 100 µg/mL GAAE for 21 h.

**Figure 2 nutrients-15-01457-f002:**
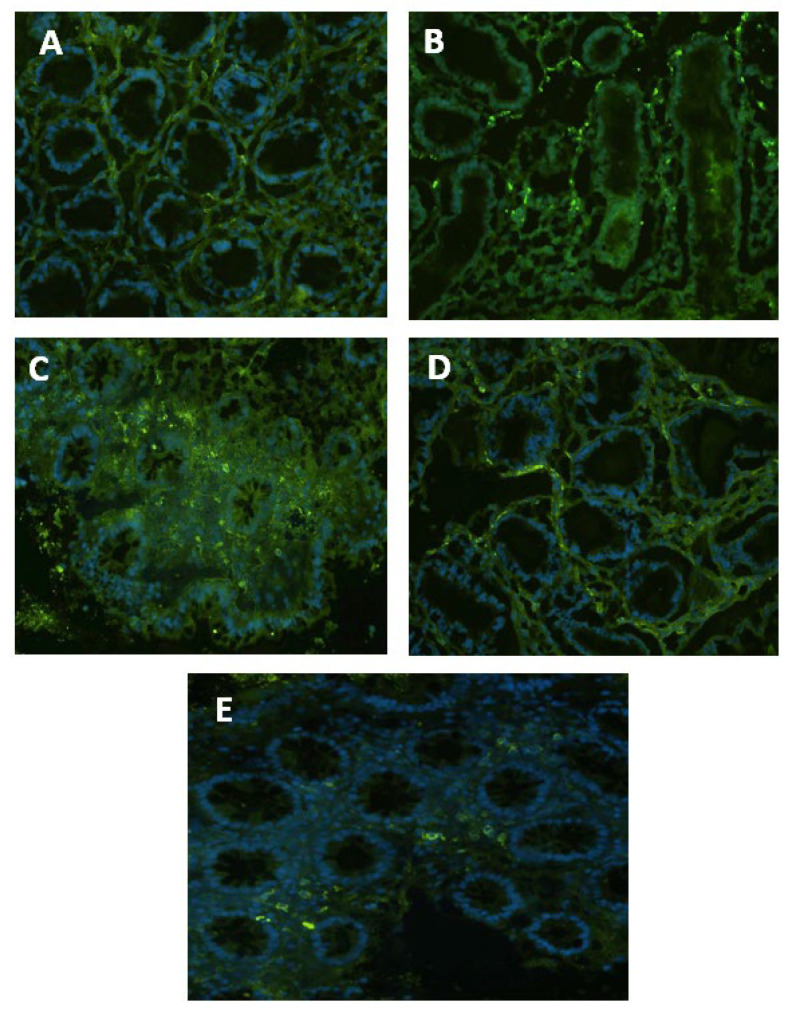
**GAAE pre-treatment effect on COX-2 expression in normal controls challenged with EC-LPS.** (**A**) No treated biopsy; (**B**) biopsy challenged with 1 µg/mL EC-LPS for 18 h; (**C**) biopsy pretreated 3 h with 50 µg/mL GAAE, then challenged with 1 µg/mL EC-LPS for 18 h; (**D**) biopsy pretreated 3 h with 100 µg/mL GAAE, then challenged with 1 µg/mL EC-LPS for 18 h; (**E**) biopsy pretreated only with 100 µg/mL GAAE for 21 h.

**Figure 3 nutrients-15-01457-f003:**
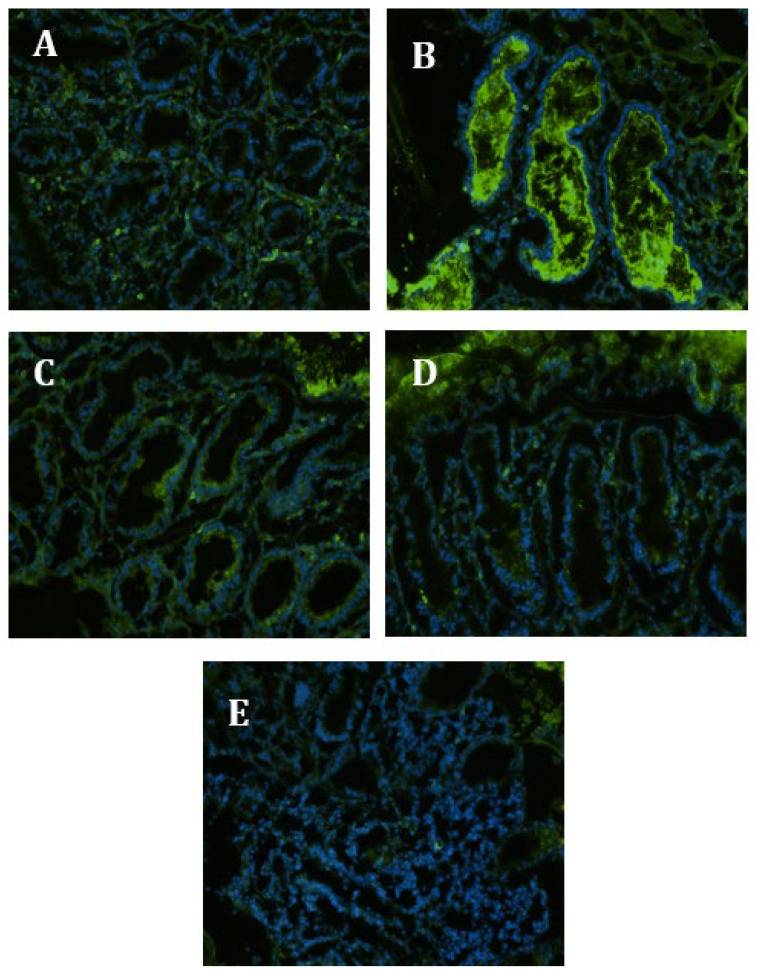
**GAAE pre-treatment effect on NF-κB expression in situ in normal controls challenged with EC-LPS**. (**A**) No treated biopsy; (**B**) biopsy challenged with 1 µg/mL EC-LPS for 18 h; (**C**) biopsy pretreated 3 h with 50 µg/mL GAAE, then challenged with 1 µg/mL EC-LPS for 18 h; (**D**) biopsy pretreated 3 h with 100 µg/mL GAAE, then challenged with 1 µg/mL EC-LPS for 18 h; (**E**) biopsy pretreated only with 100 µg/mL GAAE for 21 h.

**Figure 4 nutrients-15-01457-f004:**
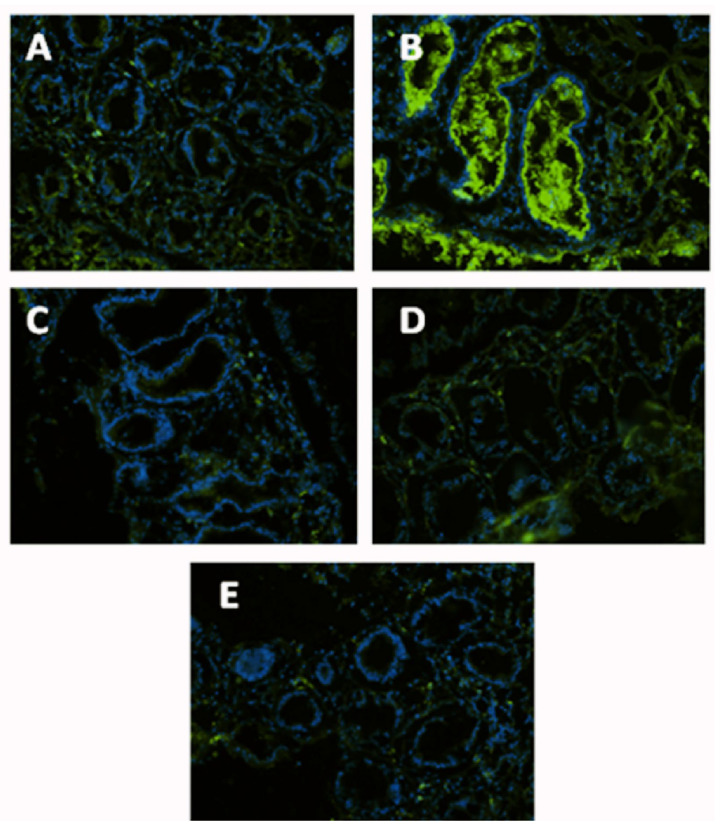
**GAAE pre-treatment effect on p38 MAPK expression in situ in normal controls challenged with EC-LPS**. (**A**) No treated biopsy; (**B**) Biopsy challenged with 1 µg/mL EC-LPS for 18 h; (**C**) Biopsy pretreated 3 h with 50 µg/mL GAAE then challenged with 1 µg/mL EC-LPS for 18 h; (**D**) Biopsy pretreated 3 h with 100 µg/mL GAAE then challenged with 1 µg/mL EC-LPS for 18 h; (**E**) Biopsy pretreated only with 100 µg/mL GAAE for 21 h.

**Figure 5 nutrients-15-01457-f005:**
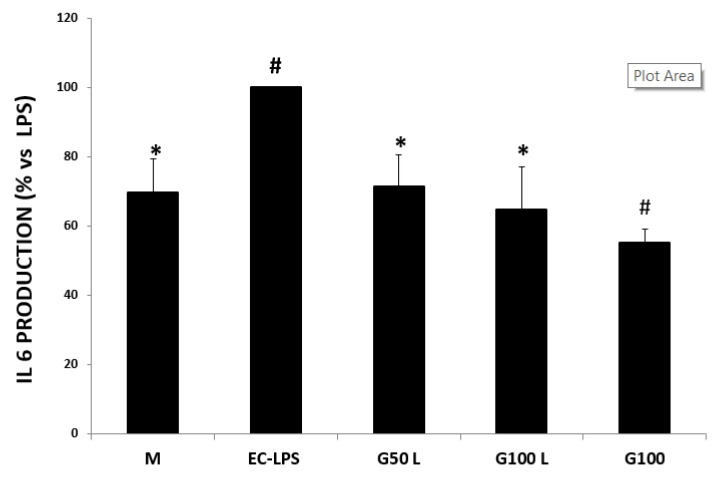
**GAAE pre-treatment effect on IL-6 production in normal controls challenged with EC-LPS**. Biopsies were treated only with medium culture (M), or with 1 mg/mL of EC-LPS or pretreated for 3 h with 50 and 100 µg/mL GAAE, then challenged for 18 h with EC-LPS (G50 L and G100 L) or treated only with 100 µg/mL GAAE (G100). Results are expressed as a percentage of each treatment with respect to EC-LPS ± S.E.M. (*N* = 15), * *p* < 0.05 vs. EC-LPS, # *p* < 0.05 vs. medium (Student’s *t*-test).

**Figure 6 nutrients-15-01457-f006:**
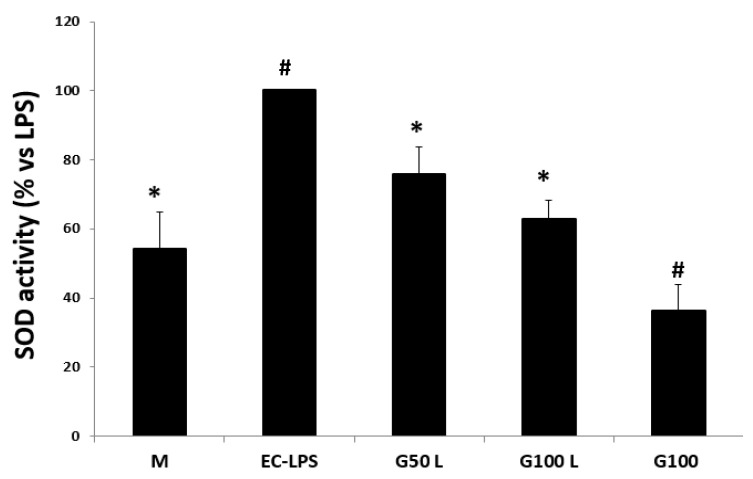
**GAAE pre-treatment effect on SOD activity in normal challenged with EC-LPS**. Biopsies were treated only with medium culture (M) or with 1 mg/mL of EC-LPS or pretreated for 3 h with 50 and 100 µg/mL GAAE, then challenged for 18 h with EC-LPS (G50 L and G100 L) or treated only with 100 µg/mL GAAE (G100). Results are expressed as a percentage of each treatment with respect to EC-LPS ± S.E.M. (*N* = 15), * *p* < 0.05 vs. EC-LPS, # *p* < 0.05 vs. medium (Student’s *t*-test).

**Figure 7 nutrients-15-01457-f007:**
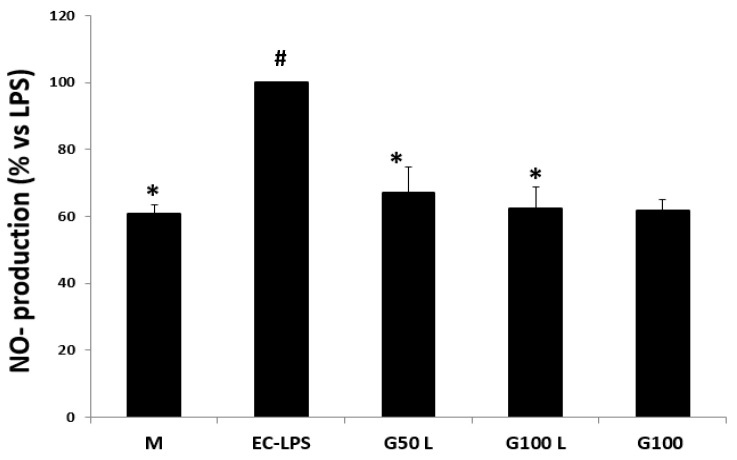
**GAAE pre-treatment effect on NO production in normal controls challenged with EC-LPS**. Biopsies were treated only with medium culture (M), or with 1 mg/mL of EC-LPS or pretreated for 3 h with 50 and 100 µg/mL GAAE, then challenged for 18 h with EC-LPS (G50 L and G100 L), or treated only with 100 µg/mL GAAE (G100). Results are expressed as means ± S.E.M. (*N* = 15), * *p* < 0.05 vs. EC-LPS, # *p* < 0.05 vs. medium (Student’s *t*-test).

**Figure 8 nutrients-15-01457-f008:**
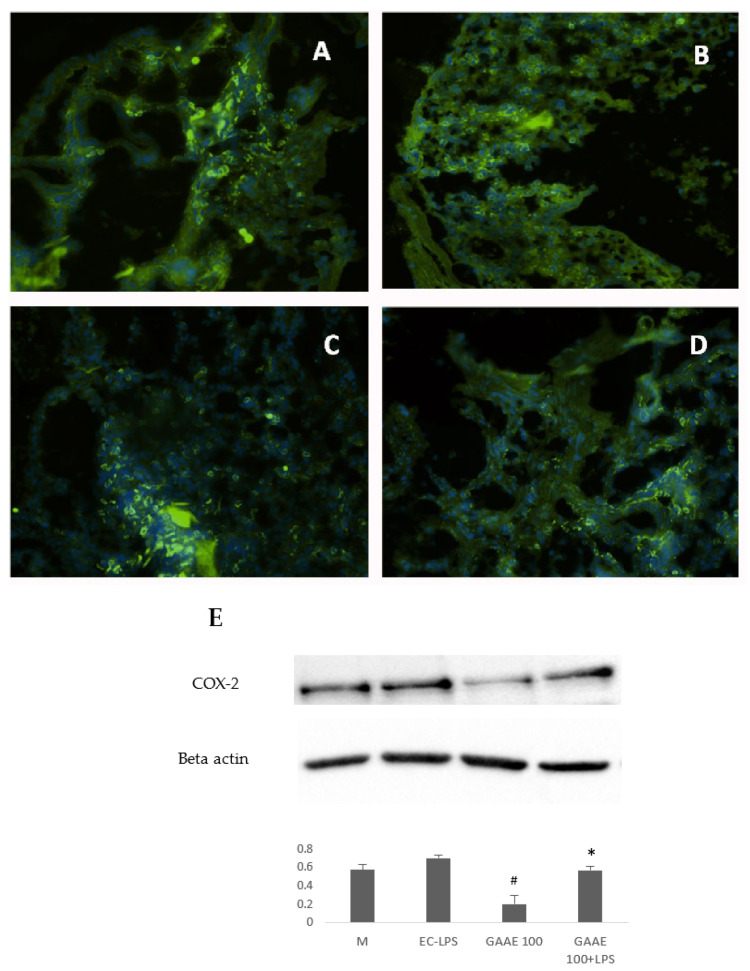
**GAAE pre-treatment effect on COX-2 expression in UC patient biopsies challenged with EC-LPS.** (**A**) Non-treated biopsy; (**B**) biopsy challenged with 1 µg/mL EC-LPS; (**C**) biopsy treated with 100 µg/mL GAAE; (**D**) biopsy co-treated with 100 µg/mL GAAE and 1 µg/mL EC-LPS; (**E**) Western blotting with the same treatments. Results are expressed as means ± S.E.M. (*N* = 31), * *p* < 0.05 vs. EC-LPS, # *p* < 0.05 vs. medium (Student’s *t*-test).

**Figure 9 nutrients-15-01457-f009:**
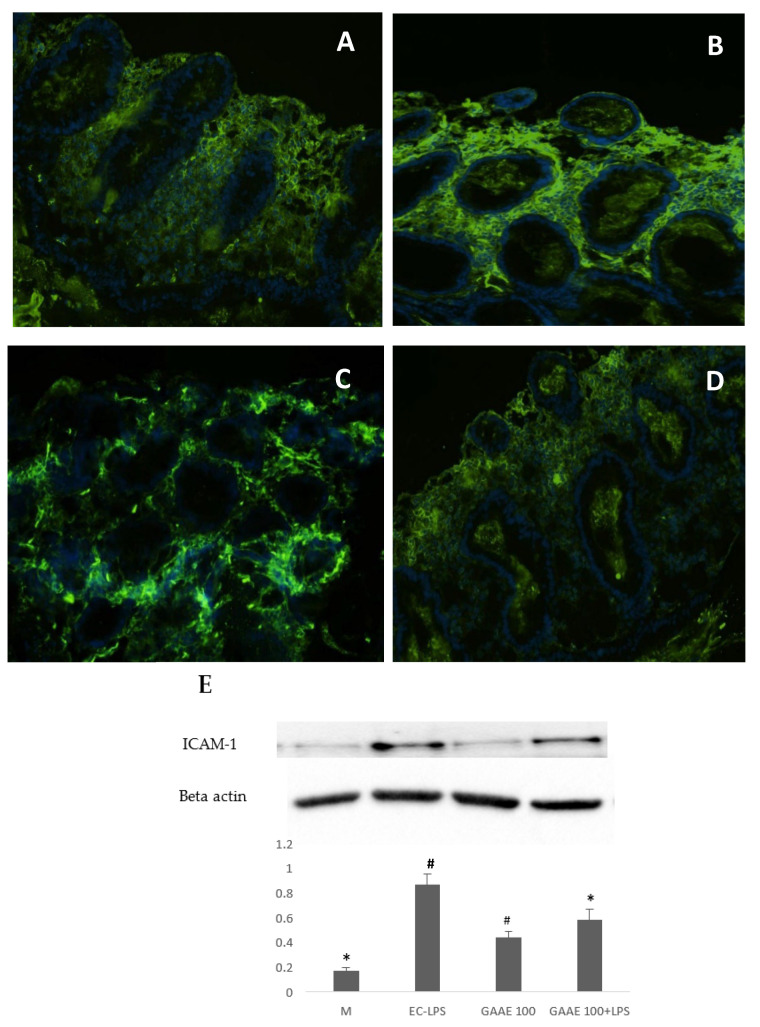
**GAAE pre-treatment effect on ICAM-1 expression in UC patient biopsies challenged with EC-LPS.** (**A**) No treated biopsy; (**B**) biopsy challenged with 1 µg/mL EC-LPS; (**C**) biopsy treated with 100 µg/mL GAAE; (**D**) biopsy cotreated with 100 µg/mL GAAE and 1 µg/mL EC-LPS; (**E**) Western blotting with the same treatments. Results are expressed as means ± S.E.M. (*N* = 31), * *p* < 0.05 vs. EC-LPS, # *p* < 0.05 vs. medium (Student’s *t*-test).

**Figure 10 nutrients-15-01457-f010:**
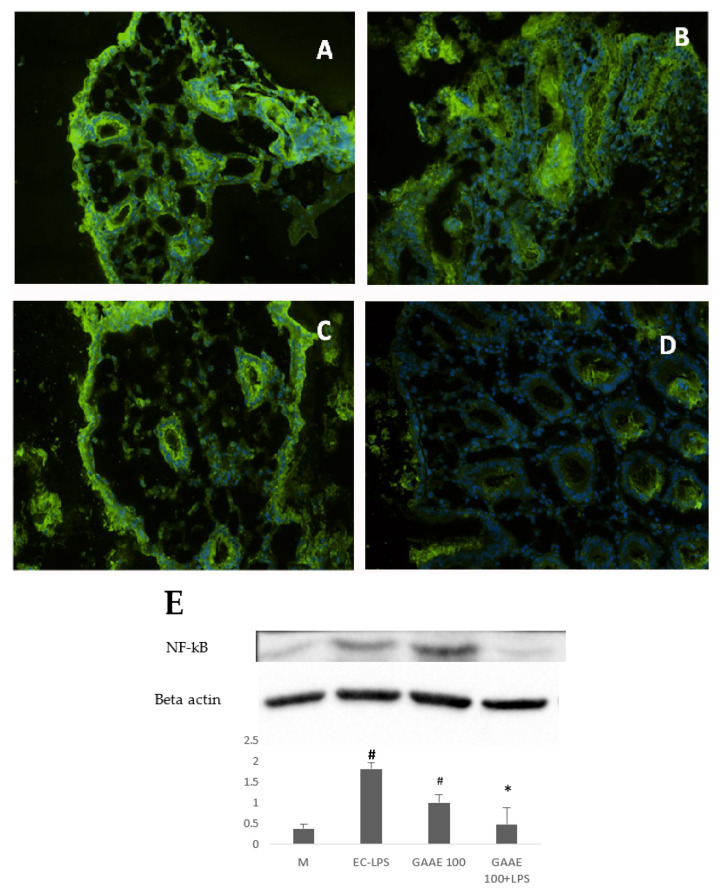
**GAAE pre-treatment effect on NF-κB expression in UC patient biopsies challenged with EC-LPS.** (**A**) No treated biopsy; (**B**) biopsy challenged with 1 µg/mL EC-LPS for 18 h; (**C**) biopsy treated with 100 µg/mL GAAE; (**D**) biopsy cotreated with 100 µg/mL GAAE and 1 µg/mL EC-LPS; (**E**) Western blotting with the same treatments. Results are expressed as means ± S.E.M. (*N* = 31), * *p* < 0.05 vs. EC-LPS, # *p* < 0.05 vs. medium (Student’s *t*-test).

**Figure 11 nutrients-15-01457-f011:**
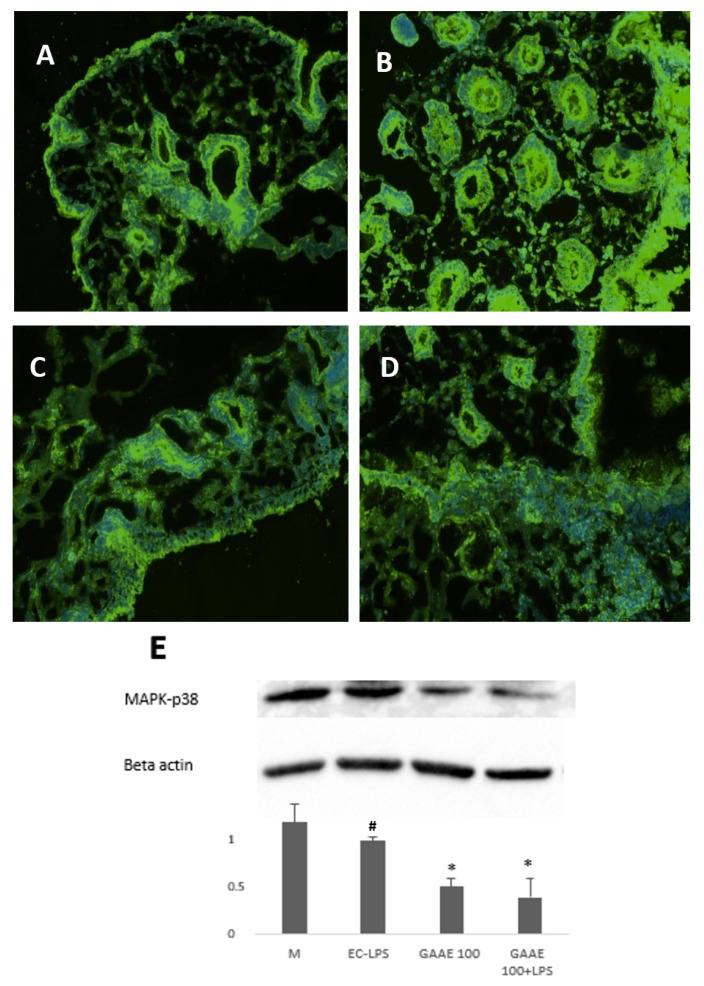
**GAAE pre-treatment effect on p38 MAPK expression in UC patient biopsies challenged with EC-LPS.** (**A**) No treated biopsy; (**B**) biopsy challenged with 1 µg/mL EC-LPS; (**C**) biopsy treated with 100 µg/mL GAAE; (**D**) biopsy cotreated with 100 µg/mL GAAE and 1 µg/mL EC-LPS; (**E**) Western blotting with the same treatments. Results are expressed as means ± S.E.M. (*N* = 31), * *p* < 0.05 vs. EC-LPS, # *p* < 0.05 vs. medium (Student’s *t*-test).

**Figure 12 nutrients-15-01457-f012:**
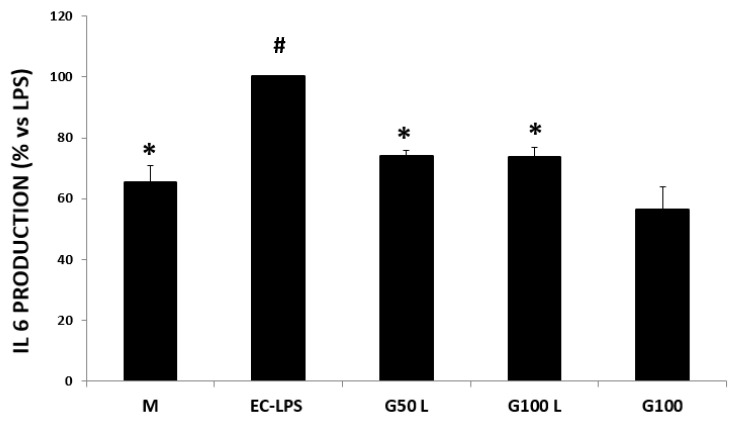
**GAAE pre-treatment effect on IL-6 production in UC patients challenged with EC-LPS**. Biopsies were treated only with medium culture (M), or with 1 mg/mL of EC-LPS or pretreated for 3 h with 50 and 100 µg/mL GAAE, then challenged for 18 h with EC-LPS (G50 L and G100 L) or treated only with 100 µg/mL GAAE (G100). Results are expressed as a percentage of each treatment with respect to EC-LPS ± S.E.M. (*N* = 31), * *p* < 0.05 vs. EC-LPS, # *p* < 0.05 vs. medium (Student’s *t*-test).

**Figure 13 nutrients-15-01457-f013:**
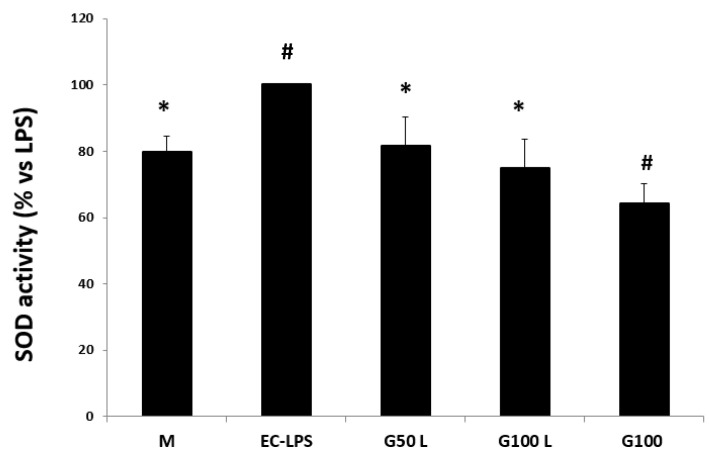
**GAAE pre-treatment effect on SOD activity in UC patients challenged with EC-LPS**. Biopsies were treated only with medium culture (M), or with 1 mg/mL of EC-LPS or pretreated for 3 h with 50 and 100 µg/mL GAAE, then challenged for 18 h with EC-LPS (G50 L and G100 L) or treated only with 100 µg/mL GAAE (G100). Results are expressed as a percentage of each treatment with respect to EC-LPS ± S.E.M. (*N* = 31), * *p* < 0.05 vs. EC-LPS, # *p* < 0.05 vs. medium (Student’s *t*-test).

**Figure 14 nutrients-15-01457-f014:**
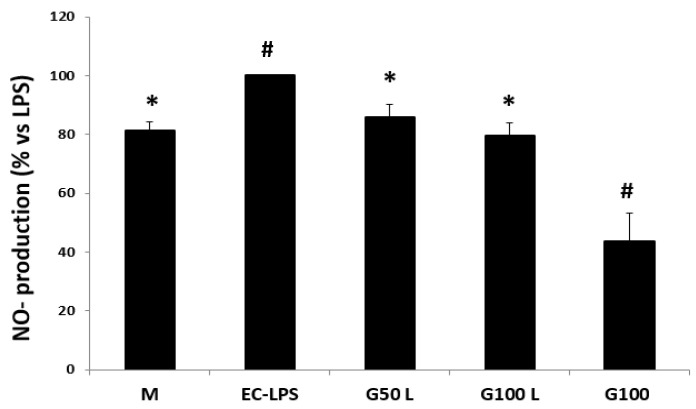
**GAAE pre-treatment effect on NO production in UC patients challenged with EC-LPS**. Biopsies were treated only with medium culture, or with 1 mg/mL of EC-LPS or pretreated for three h with 50 and 100 µg/mL GAAE, then challenged for 18 h with EC-LPS (G50 L and G100 L), or treated only with 100 µg/mL GAAE (G100). Results are expressed as means ± S.E.M. (*N* = 31), * *p* < 0.05 vs. EC-LPS, # *p* < 0.05 vs. medium (Student’s *t*-test).

## Data Availability

Data will be available upon specific, motivated request.

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
