# Peer review of "The Role of Globularia alypum Explored Ex Vivo In Vitro on Human Colon Biopsies from Ulcerative Colitis Patients"

_nutrients, 2023, doi:10.3390/nu15061457_

Round 1
Reviewer 1 Report
This manuscript by Hajji et. al. has an interesting concept of using natural extracts to attenuate inflammatory bowel disease. However, this has been already partly raised by the same authors in Transl Med UniSa. 2021; 24(1): 13–23. In their previous publication, the authors have shown the downregulation of Cox-2, ICAM-1, Mapk p38 and NF Kb in the presence of GAAE via westernblot. While the authors presently used IF, the previous findings are very similar to the current manuscript’s figures 1-4. There was also a mention of validating the findings via westernblot (pg. 7 line 195) without cross reference to a figure.
It would be more informative if there is a comparison on the effect of GAAE on normal and IBD patient-derived colon biopsies. Moreover, it would strengthen this article if they authors also measure the effect of GAAE on the classical IBD inflammatory markers such as IL-17, TNFa and IFNg.
The manuscript should be also proof read for clarity and missing data but indicated in the text (Figure 4 westernblot; Fig. 6B). The IF figures will also be easier to understand if the authors show the quantification of the staining and indicate how many samples were stained in the figure legends.
Author Response
Answers to reviewers
We thank the Editor for giving our paper the attention to be revised and considered for further publication. Then, we would thank the reviewers for this deep and serious revision which can make this work a better version ready to be published. Here you will find answers to all the comments after careful revision, modification, and replies for each one of them respectively.
Reviewer
We thank the reviewer for the considerable attention given to our work and the detailed evaluation comparison with our previous published paper. Here you find the answers for each question or comment:
- This manuscript by Hajji et al. has an interesting concept of using natural extracts to attenuate inflammatory bowel disease. However, this has already been partly raised by the same authors in Transl Med UniSa.2021; 24(1): 13–23. In their previous publication, the authors have shown the downregulation of Cox-2, ICAM-1, Mapk p38, and NF Kb in the presence of GAAE via westernblot. While the authors presently used IF, the previous findings are very similar to the current manuscript’s figures 1-4.
We really appreciate the fact that the reviewer evaluated our previous paper. They saw that the previous paper was mainly about in vitro activities of Globularia alypum. The ex vivo- in vitro study included was a primer study based on normal biopsies token from 10 persons, and the evaluation was based on 4 biomarkers by western blotting. However, the present paper presents a study based on 46 persons, including 31 ulcerative colitis patients, in which those markers were studied in situ by IF. Besides this, the effects of SOD, NO, and IL-6 were evaluated. The IF results were carried out to enhance the previous results and to see the difference in the GAAE effect on ‘normal’ and UC biopsies.
- There was also a mention of validating the findings via westernblot (pg., 7 line 195) without cross reference to a figure.
We thank the reviewer for this comment, it was our mistake. We decided not to include the western blot results of normal biopsies in this paper as they were similar to that of the previous paper you mentioned above. Still, then we forgot to modify the text. The mention of the blot has been erased in the revised version.
- It would be more informative if there is a comparison on the effect of GAAE on normal and IBD patient-derived colon biopsies.
In fact, this was one of the work purposes, and regarding to your comment we improved this in the discussion section.
- it would strengthen this article if they authors also measure the effect of GAAE on the classical IBD inflammatory markers such as IL-17, TNFa and IFNg.
The reviewer is right, as these factors are implicated in the disease. However, once evaluated the significant effect of GAAE on NF-KB, we wanted to study it more deeply by evaluating cytokines related to this cascade. For some laboratory limitations for a side, and for biopsy limitation regarding their number and size we selected Il-6 as in the bibliography it seems to be the most implicated in this activity. We mention the issues raised by the referee as limitation of the study
- The manuscript should be also proof read for clarity and missing data but indicated in the text (Figure 4 westernblot; Fig. 6B). The F figures will also be easier to understand if the authors show the quantification of the staining and indicate how many samples were stained in the figure legends.
As we mentioned in the second comment, the mention of a western blot was a mistake that we have now corrected. For the staining, the COX-2 was quantified by counting the positive cells (fluorescent cells). For the other biomarkers, the fluorescence was evaluated visually. The IF was applied in all the collected biopsy samples (15 healthy and 31 UC biopsies). We added a sentence in the Methods to clarify this point.

Reviewer 2 Report
The manuscript is in no way ready for publication, it is still only a draft and needs work. The data presented are of scientific merit but cannot be presented this way. Please see a few more specific comments. The figures are disorganized. figures 5-7 are displayed after figure 11. This is not how it should be. Figures are not to be displayed twice. In figures 5-7 there is no SD for the EC-LPS bar. Why? Were there not enough repetitions? Some figures miss letters in the figures, others in the legend. See comments in the file attached. The asticks are not centered over the bars. In the WB figures the E is in red font for some reason. Some methodological issues: Why was only IL-6 measured? Biopsies release a host of other cytokines and chemokines to the medium. The WB bands are not clear.The authors should seriously revise the manuscript and data otherwise this should be rejected.
Author Response
Answers to reviewers
We thank the Editor for giving our paper the attention to be revised and considered for further publication. Then, we would thank the reviewers for this deep and serious revision which can make this work a better version ready to be published. Here you will find answers to all the comments after careful revision, modification, and replies for each one of them respectively.
Reviewer
We want to thank the reviewer for this deep revision which will improve our paper and we hope that you find here the answers for each comment:
- The figures are disorganized. figures 5-7 are displayed after figure 1
That is true, because these figures were organized in couples for the SOD, NO and IL-6 from normal and UC biopsies to see the difference between both cases and to reduce the number of figures. However, this organization doesn’t match with the final order of the paper, so we re-organized the numbers regarding to their positions in the text.
- In figures 5-7 there is no SD for the EC-LPS bar
In fact, the results are presented on percentage in which the EC-LPS was the 100% that why there is no SD for it.
- Some figures miss letters in the figures, others in the legend. The sticks are not centered over the bar.
Thank you, we admit a shabby presentation of our work. As you can check in the manuscript, we have adjusted those points as well.
- In the WB figures the E is in red font for some reason
The red color was used just to differentiate this letter from the figures as they were all in black, but we modified them to be black as well.
- Why was only IL-6 measured? Biopsies release a host of other cytokines and chemokines to the medium. The B bands are not clear.
The reviewer is right but, we have analyzed the SOD and NO in the same mediums with different repetitions. We had to choose what cytokine measure. Due to the biopsies' small size, the culture medium's remaining quantity didn’t allow us to run a larger number of tests. Therefore, we choose the IL-6 as it seems one of the most related to the NF-KB activity along the literature.
Moreover, the small size of biopsies gave a low quantity of these biomarkers which made WB bands less intense. We mention the issues raised by the referee as limitations of the study.

Reviewer 3 Report
1. English should be improved - same sentences should be re-write nad change since their not correct in English
2. Diasadvantage:
- small study group
- why only UC? it wasn’t explained
- why healthy Controls had colonoscopy?
- too many figures
3. advantages
- novelty
- comprehensive presentation
Author Response
Answers to reviewers
We thank the Editor for giving our paper the attention to be revised and considered for further publication. Then, we would thank the reviewers for this deep and serious revision which can make this work a better version ready to be published. Here you will find answers to all the comments after careful revision, modification, and replies for each one of them respectively.
Reviewer
We want to thank the reviewer for this significant revision. Below are answers for your comments:
- English should be improved - same sentences should be re-write nad change since their not correct in English
We have revised all the manuscript and hopefully improved the English.
- - small study group
This study includes a number of 46 persons. The total group number was larger, but it included some Crohn’s patients for which the number was not enough to evaluate the results, besides several non-valid results related to different reasons, as the presence of some patients already under other biological therapies. The Italian long Covid-19 period was also a big limitation to get a higher number of samples. We mention that issue as a limitation of the study in Discussion
- why only UC? it wasn’t explained
The study protocol also included CD patients, but as the number of eligible patients suffering of colonic CD was small, not enough to make comparisons. Therefore, we decided to focus on UC. Moreover, the few results got from CD were different from those of UC, and that was going to increase the final number of figures, results, and their evaluation.
- why healthy Controls had colonoscopy?
Patients sent for colonoscopy come under the request of their doctors for IBS or blood loss from hemorrhoids. Many of those patients are found with healthy colons, from which we took the biopsies to be compared to inflamed ones.
- too many figures
In fact, we agree. We tried to minimize the number of figures by combining some results like IF with WB in one figure, and we tried to limit the study to UC with the most relevant biomarkers. However, it is difficult to show results as numbers in the text as it limits the readability of the paper. In the revised text we better organized the figures.

Round 2
Reviewer 1 Report
The authors have improved the manuscript by including new data (figures 8-14) based the UC biopsies. However, the it is still very hard to derive a conclusion based on IF pics, without the statistics from all stained samples and markers. In some cases, the authors were quantifying the IF+ cells, this was not consistently done in all figures.
To see the difference in the GAAE effect on healthy and UC biopsies, the data should be presented and compared together.
The new westernblot analysis for NFkb and Mapkp38 does not really support activation the activation of the signalling pathway. Phosphorylation of these markers should be shown.
Lastly, the figures and figure legends are very sloppily presented. Some figures were clearly cropped and missing axis. Not all legends have the full description of the experiment done (ie. Number of experimental replicates and statistical analysis).
Author Response
The authors thank the reviewer for his ongoing contributions to the paper's development.
-The authors have improved the manuscript by including new data (figures 8-14) based the UC biopsies. However, it is still very hard to derive a conclusion based on IF pics, without the statistics from all stained samples and markers. In some cases, the authors were quantifying the IF+ cells, this was only consistently done in some figures.
No additional data were added to the paper. However, as recommended by the reviewers, we did arrange the results better, We changed the numbering of the figures since they were combined initially as A and B of the control and UC outcomes of each marker, then split for clarity. The reviewer is right, some of the stainings do not have figures and statistics. The IF for COX-2 allowed us to count the positive cells. For the other markers, the fluorescence was not localized in the cells, so it could be counted and evaluated by the semiquantitative method currently used in most labs,
We acknowledge, however, that the presentation of figures is important, and we added a number of tables in Supplementary material , including all the markers with the positive cells for COX-2 and the intensity evaluation of the other ones expressed as intensity from 0 absent to 3 high.
To see the difference in the GAAE effect on healthy and UC biopsies, the data should be presented and compared together.
The data were initially organized in this way. Still, we chose to reorganize and separate them for many reasons, including the many figures and the exclusion of western blotting in controls. In the present version, with the addition of the tables, the availability of data and comparison is improved .
-The new westernblot analysis for NFkb and Mapkp38 does not really support activation the activation of the signalling pathway. Phosphorylation of these markers should be shown.
Lastly, the figures and figure legends are very sloppily presented. Some figures were clearly cropped and missing axis. Not all legends have the full description of the experiment done (ie. Number of experimental replicates and statistical analysis).
As previously stated, no new results were added; nonetheless, the reviewer is correct concerning the benefits of adding phosphorylation. Regrettably, we were unable to conduct more experiments due to the paucity of tissue of all patients and controls. Moreover, our results with figures are already too extensive to include any more. Thank you also for pointing up the erroneous figures. In truth, most of those issues stemmed from the paper's form adjustment and the hefty figures, which changed with each modification, resulting in the loss of some parts of the figures. We attempted to edit them and turned some of them into images to avoid future alterations. We also supplied the missing information from the experiment description.

Reviewer 2 Report
See attached. This is much improved but there are still text issues. Some may be a result of file conversion. Please see comments in the attached file.

Author Response
We want to express our appreciation to the reviewer for his second revision. The reviewer is correct; several errors were caused during the file conversion, and we attempted to alter them differently in order to avoid this problem. In the experiment description, we included the missing letter D. Also, we found that two author names were highlighted. These names were accidentally left out of the submitted version even though we had entered them into the system prior to submission, so we fixed this error in the article.
